# Fabrication of a New Electrochemical Sensor Based on Bimetal Oxide for the Detection of Furazolidone in Biological Samples

**DOI:** 10.3390/mi13060876

**Published:** 2022-05-31

**Authors:** Ruspika Sundaresan, Vinitha Mariyappan, Shen-Ming Chen, Saranvignesh Alagarsamy, Muthumariappan Akilarasan

**Affiliations:** Department of Chemical Engineering and Biotechnology, National Taipei University of Technology, No. 1, Section 3, Chung-Hsiao East Road, Taipei 106, Taiwan; t109a09415@ntut.edu.tw (R.S.); t108519401@ntut.edu.tw (V.M.); t110a09405@ntut.edu.tw (S.A.); akilarasan@ntut.edu.tw (M.A.)

**Keywords:** bimetal oxide, hydrothermal, furazolidone, glassy carbon electrode, cyclic voltammetry

## Abstract

This study utilized a simple hydrothermal method to synthesize nickel molybdenum oxide (NMO) for the detection of furazolidone (FZE). Our synthesized NMO was characterized by X-ray diffraction (XRD), X-ray photoelectron spectroscopy (XPS), Raman spectroscopy, Fourier transform infrared spectroscopy (FTIR), field emission scanning electron spectroscopy (FE-SEM), and energy dispersive X-ray spectroscopy (EDX). The cyclic voltammetry (CV) and differential pulse voltammetry (DPV) were used to detect the FZE. Under optimized conditions, the obtained results showed that the NMO had an excellent electrocatalytic property towards FZE. As a result, NMO/GCE showed a good linear range of 0.001–1765 µM, an excellent detection limit (LOD) of 0.02 µM, and sensitivity of 0.2042 µA µM^−1^ cm^−2^.

## 1. Introduction

Furazolidone (FZE), namely, 3-(5-nitrofurfurylideneamino)-2-oxazolidinone, is a synthetic broad-spectrum antibiotic with a nitro group in its structure [1]. It is a chemotherapeutic drug, and because of its low cost of manufacture and high efficacy, it has also been used as a growth stimulant and additive in food-producing animals. For humans, it is mainly used in treating diarrhea and cholera. However, medical research in recent years confirms that FZE is carcinogenic to human health [1]. However, the European Union was unable to assign a maximum residue limit for FZE because of its carcinogenic effects on human health. As a result, for human health, FZE usage was banned in the European Union, Brazil, Philippines, Thailand, and Australia. As for its low price and effective promotions in poultry, FZE is used illegally in many countries [2]. Hence, a trace amount of FZE detected in biological and water samples become essential. Spectrophotometry [3], high-performance liquid chromatography [4], chemiluminescence [5], and electrospray ionization mass spectrometry [6] have been employed to detect FZE so far. Despite high accuracy and good sensitivity, these methods require sophisticated equipment at a high cost. Therefore, electrochemical sensors are more advantageous due to their low cost, rapid response, user-friendly and on-site detection. Recently, Bao-Shan He et al. modified graphene and Au nanoparticles [1], Selvarasu Maheshwaran et al. developed 1-Dimensional (1D)—2-Dimensional (2D) BiVO_4_@MoS_2_ [7], and Chellapandi et al. developed a 3D hierarchical CuS-rGO/g-C_3_N_4_ nanostructure [8] for the electrochemical detection of FZE.

Among metal oxides, binary metal oxide materials have attracted great interest in developing active electrode materials because of their feasible oxidation state and electrical conductivity [9]. Recently, transition series elements have been found as a promising material in the fields of supercapacitors [9], photodegradation [10], batteries [11], water splitting [12], and sensors [13]. As metal molybdates, MMoO_4_ (M = Mn, Co, Zn, Fe, Ni, etc.) have high electrical conductivity and specific activity, and they have emerged as promising advanced electrode materials in electrochemical applications [14]. NiMoO_4_ (NMO) is an efficient electrode material due to its low cost, chemical stability, strong electrochemical activity of Ni, and conductivity of Mo.

In this research, we successfully prepared NMO rods by a cost-effective facile hydrothermal method. The prepared NMO rods were used for FZE detection. The proposed sensor exhibited characteristics such as a low detection limit, wide linear range, high sensitivity, anti-interference ability, and long-term stability. As far as we know, this is the first work employing NMO rods for the electrochemical detection of FZE.

## 2. Experiments

### 2.1. Reagents

Nickel (II) chloride hexahydrate (NiCl_2_.6H_2_O,99.9%), ammonium heptamolybdate tetrahydrate ((NH_4_)_6_Mo_7_O_24_·4H_2_O), sodium phosphate monobasic dihydrate (NaH_2_PO_4_·2H_2_O, ≥99.0%), sodium phosphate dibasic (Na_2_HPO_4_, ≥99.0%), potassium ferrocyanide (K_4_ [Fe(CN)_6_]), potassium ferricyanide (K_3_ [Fe(CN)_6_]), potassium chloride (KCl, 99.0–100.5%), and hydrochloric acid (HCl, 36.5–38%) were purchased from Sigma Aldrich and used without further purification. Deionized (DI) water and ethanol were used for solution making and other washing purposes throughout the experiments. All experiments were carried out in N_2_ gas saturated with 0.1 M phosphate buffer solution (PBS).

### 2.2. Material Characterization

The morphology and the structure of the as-synthesized material were analyzed by using field emission electron microscopy (FESEM, Hitachi S-3000). The XRD pattern of the material was recorded (XRD, X’Pert-PRO, PANalytical B.V., and The Netherlands). The composition and chemical state of the catalyst was analyzed by X-ray photoelectron spectroscopy (XPS, Thermo ESCALAB 250). The Raman spectrum data were obtained from a Micro-Raman spectrometer (Raman Dong Woo 500 I, Korea). FT-IR spectroscopy (Perkin-Elmer IR spectrometer) was used to detect the functional groups. The electrochemical measurements were verified by using a three-electrode system consisting of a glassy carbon electrode (surface area 0.071 cm^2^) as a working electrode, platinum wire as a counter electrode, and Ag/AgCl as a reference electrode. The electrochemical experiments were carried out in an electrochemical workstation with cyclic voltammetry (CV CH1205C) and differential pulse voltammetry (DPV CHI900) in the potential range of +0.5 to +1.2 (Ag/AgCl vs. V) at a scan rate of 50 mVs^−1^. A Nyquist plot was obtained by electron impedance spectroscopy (EIS).

### 2.3. Synthesis of NMO Rods

To synthesize the NMO rods, 0.5 g of NiCl_2_.6H_2_O and 0.5 g of (NH_4_)_6_Mo_7_O_24_·4H_2_O were stirred for about 30 min in 60 mL of DI water. The mixture was transferred to a 80 mL Teflon-lined stainless steel autoclave, and kept for a period of 6 h at a temperature of 150 °C in a muffle furnace [15]. After cooling, the obtained product was washed several times with water and ethanol and dried in an oven at 50 °C for 24 h. Hence, the powder was ground and named NMO rods, which can be used for further electrochemical reaction and characterization. The schematic representation of as-synthesized NMO is shown in Figure 1.

### 2.4. Fabrication of Electrodes

Initially, NMO was dissolved in DI water and sonicated for 30 min to disperse the material. Subsequently, the GCE was cleaned with alumina slurry and then washed with DI water and ethanol several times. The NMO rods were fabricated over the GCE for the detection of FZE; 6 µL of NMO was drop-casted over the cleaned GCE and dried in an oven at 50 °C. The modified electrode was named NMO/GCE.

### 2.5. Choice of Material

NMO is considered a potential electrode material because of its low cost, high stability, high specific capacitance, low toxicity, and high abundance. Owing to these advantages, we fabricated an NMO/GCE electrode for the detection of FZE. NMO/GCE enhances the electrochemical parameters such as stability, selectivity, repeatability, and reproducibility.

## 3. Result and Discussion

### 3.1. Structure and Morphology of NMO Rods

XRD analysis was used to scrutinize the crystallinity of the as-synthesized NMO rods. Figure 1a depicts the NMO rods diffraction at 2Ɵ values of 14.3°, 21.1°, 23.6°, 27.4°, 29.7°, 31.0°, 33.1°, 35.7°, 40.2°, 43.3°, 45.3°, 48.8°, 54.8°, 58.0°, and 64.6°, which correspond to the planes of (110), (111), (021), (−221), (−311), (130), (−131), (−203), (040), (−421), (113), (−512), (133), (−152), and (−244), respectively. The obtained peaks were well-matched with NiMoO_4_ JCPDS NO. 01-086-0361 [16], which demonstrates the successful formation of NMO rods. The obtained narrow peaks indicate that our synthesized NMO rods have high crystallinity and purity.

The Raman spectrum of NMO rods is displayed in Figure 1b, which exhibited a prominent peak at 273, 367, 833, 879, and 954 cm^−1^. The peaks at 833, 879, and 954 were attributed to the asymmetric and symmetric stretching modes of Mo-O bonds, and the peaks at 367 and 273 cm^−1^ were obtained due to the bending mode of Mo-O and deformation modes of Mo-O-Mo, respectively [17]. Figure 1c shows the FTIR spectrum of NMO rods, which exhibited peaks at 3381, 1614, 964, 744, and 448 cm^−1^; the obtained result well-matches that in a previous report [17]. The distorted MoO_4_ tetrahedron υ_1_ and υ_2_ vibrations occurring in NMO were attributed to the peaks at 967 and 744 cm^−1^. The peak positioned at 448 was assigned to the υ_4_ and υ_5_ modes of MoO and υ_3_ modes of NiO_6_ of NMO rods. The peaks positioned at 3440 and 1624 cm^−1^ correspond to the flexing modes and stretching vibration of O–H originating from the absorbed water of the NMO rod surfaces.

The chemical composition of as-synthesized NMO was examined by XPS. The overall survey spectrum confirms the existence of Ni, Mo, and O, which are displayed in Figure 2a. The Ni 2p spectra exhibited two peaks at 855.9 and 874.7 that were ascribed to 2p_3/2_ and 2p_1/2_ depicted in Figure 2b. The peaks at 862.9 and 877.5 eV were assigned to satellite peaks, which correspond to the characteristics of Ni^2+^ [18]. The peaks at 235.7 and 232.6 eV related to Mo 3d are shown in Figure 2c with respect to 3d_3/2_ and 3d_5/2_, and excited Mo has a +6 oxidation state [19]. The XPS spectra of O 1s are shown in Figure 2d, which displays a predominant peak at 530.1 that corresponds to the metal–oxygen bond [20].

The morphology of the synthesized NMO was investigated by FE-SEM (field emission scanning electron microscopy). Figure 3a shows the FE-SEM images of NMO, which indicates that our hydrothermally prepared NMO had a rod-like structure and was less aggregated. Figure 3b–e displays the magnified FE-SEM images of NMO, which revealed that NMO rods were irregularly arranged and the rods had rough surfaces. Figure 4a–d displays the EDX mapping of NMO, which confirms the existence of nickel (Ni), molybdenum (Mo), and oxygen (O). The other peak belongs to indium due to the ITO plate [21]. Additionally, Figure 4e shows the EDX elemental spectra of NMO with a weight percentage of 32.3%, 33.3%, and 34.2%. These results confirm the successful formation of NMO rods.

### 3.2. Electrochemical Impedance Study

The interfacial properties of the electrode were investigated by an electrochemical impedance study. The surface of the bare GCE and NMO/GCE was investigated by the EIS technique. The Nyquist plots of bare GCE and NMO/GCE are shown in Figure 5a, which were recorded in 5 mM [Fe(CN)_6_]^3−/4−^ with 0.1 M KCl solution. At high frequencies, the surface electron transfer resistance equals the semicircle formed partially. The charge transfer resistance (R_ct_) values were calculated to be 444.4 and 299 Ω for bare GCE and NMO/GCE, respectively. The observed low R_ct_ value indicates the high electrical conductivity of NMO/GCE compared to bare GCE, which confirms that the NMO modified GCE has more active sites compared to bare GCE.

The CV technique was used to examine the catalytic activity of NMO/GCE in a ferric/ferrous system containing 0.1 M KCl solution at a scan rate of 50 mVs^−1^ shown in Figure 5b. The NMO/GCE had a higher redox peak current compared to bare GCE. The excellent electrocatalytic activity of NMO/GCE is due to the large surface area and increased number of active sites. The peak-to-peak separation value of bare GCE and NMO/GCE was 0.147 and 0.184 mV, respectively. The highest redox peak current and low peak-to-peak separation of NMO/GCE are due to the NMO rod structure, large surface area, and more active sites; therefore, NMO/GCE is exposed to quick electron transfer in a Ferri/Ferro system.

The electroactive surface area of bare GCE and NMO/GCE was calculated by different scan rate investigations by CV analysis. Figure 5c shows the CV redox current of NMO/GCE at various scan rates of 20–300 mVs^−1^ in a [Fe (CN)_6_]^3−/4−^ system containing 0.1 M KCl solution. With increasing scan rate, the redox peak current increased, which reveals the fast kinetics of NMO/GCE. Figure 5d displays the linear plot of the redox current vs. square root of the scan rate and a linear regression equation with co-efficient value for anodic and cathodic I_pa_ = 0.0003x + 7 × 10^−6^, R^2^ = 0.9994 and I_pc_ = −0.0002x − 2 × 10^−5^, R^2^ = 0.9965, respectively. The Randles–Sevcik equation was used to calculate the electroactive surface area of NMO/GCE,
I_pa_ = (2.69 × 10^5^) *n*
^3/2^ ACD ^1/2^
*v*
^1/2^(1)
where I_pa_ is peak current value, *n* is number of electrons involved in the electrochemical reaction, Dis the diffusion coefficient, *v* is scan rate, and C is concentration of the Ferro ferricyanide solution. The calculated EASA of bare GCE and NMO/GCE was 0.72, and 0.117 cm^2^, respectively. The results revealed that NMO/GCE had a higher EASA value, which will be more advantageous for FZE detection.

### 3.3. Electrochemical Behavior of GCE and Modified GCE

For efficient electrochemical determination, the loading of the catalyst is a key parameter. The surface of the electrode was coated with different loading concentrations with FZE (100 µM) at a scan rate of 50 mVs^−1^, depicted in Figure 6a. According to the results, the current response of 6 µL shows the highest peak current. Its corresponding bar diagram for the CV response current vs. loading of the catalyst is shown in Figure 6b. Thus, 6 µL concentration was used for the further electrochemical detection of FZE.

The electrochemical behavior of bare GCE and NMO/GCE was investigated by CV analysis for the detection of FZE, shown in Figure 6c. The electrochemical current response of NMO/GCE and bare GCE was recorded at a scan rate of 50 mVs^−1^ containing 100 µM of FZE. Figure 6d shows the bar graph of the CV current response of modified electrodes. The reduction current response of NMO/GCE was higher than the bare GCE because NMO has a large surface area and more active sites; therefore, it exhibited good electrocatalytic activity and fast electron transfer.

The pH range of electrolytes plays a major role in the electrochemical response of the NMO/GCE. The effect of pH on NMO/GCE was evaluated by CV analysis at a scan rate of 50 mVs^−1^ containing 100 µM of FZE (pH-7), shown in Figure 7a. The cathodic peak current moved towards more negative potential and the current increased as the pH increased from 3 to 7 and decreased above pH 7 due to a lack of protons. The calibration plot for the pH vs. current and potential is shown in Figure 7b, which shows a linear regression equation and co-efficient value of I_pc_ = −0.0395x − 0.1355 and R^2^ = 0.9989, respectively. According to the results, pH 7 was chosen as a supporting electrolyte for the electrochemical detection of FZE. The reduction of a nitro group (–NO_2_) in FZE to hydroxylamine (–NHOH) is shown in Figure 2.

The effect of concentrations of FZE at NMO/GCE was observed by the CV analysis, and the CV current response of FZE (20–100 µM) at NMO/GCE was performed at a scan rate of 50 mVs^−1^, shown in Figure 8a. From the figure, it was clear that as the FZE concentration increased, the reduction peak current also increased. A linear relationship between the FZE concentrations and the peak current is shown in Figure 8b, which displays a linear regression equation and co-efficient value of I_pc_ = −0.3195x − 2.75 and R^2^ = 0.9941, respectively. According to the results, NMO/GCE has great electrocatalytic activity and fast electron transfer for the detection of FZE.

The kinetic electrochemical performance of NMO/GCE was examined at various scan rates with 100 µM of FZE at a scan rate of 20−100 mVs^−1^, shown in Figure 8c. Based on the observation, when the scan rate increased, the reduction peak current also increased. Additionally, Figure 8d shows the linear plot for the reduction peak current and the square root of scan rates and displays a linear regression equation and co-efficient value of I_pc_ = −107.9x − 14.411 and R^2^ = 0.9945, respectively.

### 3.4. Detection of FZE

The electro-catalytic activity of NMO/GCE towards FZE was examined using DPV analysis. Figure 9a shows the DPV current response of NMO/GCE, which was performed in 0.1 M PB (pH-7) with increasing FZE concentrations. When increasing the concentrations of FZE from 0.001–1765 µM, the reduction peak current also increased. The linear plot of the reduction peak current versus concentrations of FZE is shown in Figure 9b and shows a linear regression equation with a coefficient value for low and high concentrations of FZE as I_pc_ = −0.1324x − 5.2921, R^2^ = 0.9976, and I_pc_ = −0.0145x − 17.07, R^2^ = 0.9989, respectively. The LOD was determined to be 0.022 µM, calculated using the standard equation formula LOD = 3 s/q, where “s” is the standard deviation of the blank line and “q” is the slope of the calibration plot (Figure 9b). The sensitivity was calculated to be 0.2042 µA µM^−1^ cm^−2^. The electrochemical determination of NMO/GCE towards FZE was related to the results of the other electrodes shown in Table 1. Thus, the above results confirm that NMO/GCE has a wide linear range and a low LOD for the detection of FZE.

### 3.5. Selectivity, Repeatability, Reproducibility, Long-Term Stability Studies, and Real Sample Analysis of FZE Using NMO/GCE

The DPV method is used for evaluating the selectivity of the NMO/GCE sensor, which was performed in the presence of FZE (100 µM) along with some interfering molecules and ions such as (2) glucose, (3) ascorbic acid, (4) dopamine, (5) 4-nitrophenol, (6) urea, (7) K^+^, (8) Na^+^, (9) Mg^+^, (10) Cl^−^, and (11) Co^2+^, which were added, and the DPV response is displayed in Figure 8c. According to the result, the peak current of FZE was not affected even after the addition of 100-fold excess concentrations of interfering molecules and ions, which confirms that our proposed NMO/GCE sensor has strong selectivity towards FZE. Figure 8d displays the corresponding bar graph diagram of the DPV current response and interferents.

The repeatability of FZE (100 µM) on the same NMO/GCE was recorded in five repeatable measurements. Figure 10a displays the corresponding bar graph diagram for five repeatable measurements of NMO/GCE, and RSD was calculated to be 3.2%, which reveals that the NMO/GCE electrode has excellent repeatability. The reproducibility was investigated using five different modified NMO/GCE electrodes with the addition of 100 µM of FZE, and Figure 10b displays the corresponding bar graph diagram for five different modified GCEs, which reveals the better reproducibility with an RSD value of 3.0%, confirming that our proposed sensor has good reproducibility. Further, the prolonged stability of NMO/GCE was evaluated with the addition of 100 µM of FZE for 15 days. The corresponding bar graph diagram of the peak current is displayed in Figure 10c. The experiment was performed every day. Here, the electrode was maintained at 5 °C in a refrigerator. The NMO/GCE electrode retained 96.7% of the current on that first day, which shows excellent stability for NMO/GCE for the detection of FZE.

The feasibility of NMO/GCE was evaluated in human serum and urine by DPV analysis. The human serum and urine were collected from healthy volunteers. The collected biological samples were filtered by centrifuging for 20 min at 6000 rpm. The collected filtrates were diluted with 0.1 M PB. Moreover, a known concentration of FZE (5, 10, 15 µM) was spiked into the human serum and urine samples, and calculated using the standard addition method. The recovery results are shown in Table 2. The recoveries ranged from 99.6 to 98.5%. The obtained recovery results established that our proposed sensor can be used as an accurate and efficient sensing platform for the detection of FZE in real samples.

## 4. Conclusions

In summary, we synthesized NMO/GCE via the hydrothermal method as an electrochemical sensor to detect FZE. Experimental results showed that NMO/GCE had excellent electrocatalytic activity towards FZE, possessing a broad linear range of 0.001–1765 µM, narrow LOD of 0.02 µM, selectivity, reproducibility, repeatability, and stability. The modified GCE was employed for the electrochemical detection of FZE. Due to rapidity, high sensitivity, and low cost, the proposed electrochemical sensor NMO/GCE provides a promising future for sensitive and simple detection of FZE in biological samples.

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
