# Peer review of "Fabrication of a New Electrochemical Sensor Based on Bimetal Oxide for the Detection of Furazolidone in Biological Samples"

_micromachines, 2022, doi:10.3390/mi13060876_

Round 1

Reviewer 1 Report

The manuscript reports the preparation and use of nickel molybdenum oxide (NMO) rods as a sensing material for furazolidone detection. In general, the manuscript is very well written and the material characterization is very well performed with appropriate methods. However, a clear novelty in comparison to previous reports and some important discussions are still missing from the manuscript. Therefore, a revision is suggested.

The following need revision:

- The authors must clearly demonstrate that the proposed material leads to some advantages over other existing materials. Since this is a key point, it must be addressed.

- The authors also should provide a better explanation of the function of NMO. As it stands, the manuscript is purely descriptive but does not give enough information and an explanation of why this choice of material has resulted in the best solution.

- Table 1 and the context the authors provided are rather coarse. The authors do not adequately explain why the reader of this article should use the present material as well as the present method rather than the other existing materials and protocols that possibly work quite well. I suggest that the authors add more specific features of their work, especially the preparation protocol and steps in comparison to others in Table 1. Besides excellent LOD and wide linear range, more explanation of why the reader should use the protocol presented in this work rather than the others should be discussed in detail within the context.

- I recommend the authors move the figures and the experimental detail from the supporting material to the main text in the manuscript.

- In Fig.5(a) and (b), the authors only stated the loading amount in volume. The authors should provide the concentration of material used in the text.

- For the selectivity, the authors should explain why 4-nitrophenol does not show any effect on furazolidone detection as it also contains the nitro group.

- The authors mentioned that the calculated EASA of NMO/GCE was higher than bare GCE but the calculated values are not in agreement with the explanation.

- Typos and grammatical errors within the manuscript have to be checked.

Author Response

Reviewer comments and Author’s response

Journal: Micromachines

Manuscript Number:  micromachines-1749731

Authors: Sundaresan Ruspikaa, Vinitha Mariyappana, Shen-Ming Chena,*, Saranvignesh Alagarsamya, Muthumariappan Akilarasana.

Title: Fabrication of a new electrochemical sensor based on bimetal oxide for the detection of furazolidone in biological samples.

Dear Sir/Madam,

We would like to express our sincere gratitude for the valuable suggestions from reviewers to improve the quality of our manuscript. We made all the corrections according to the comments and highlighted them in pink color. The following is the list of amendments and points addressing the comments for your report.

#Reviewer 1

  1. The authors must clearly demonstrate that the proposed material leads to some advantages over other existing materials. Since this is a key point, it must be addressed.

Response: We thank for the reviewer’s comment, as per the comment we added the information about the advantages of material among the other existing material was given in choice of material and table 1.

  1. The authors also should provide a better explanation of the function of NMO. As it stands, the manuscript is purely descriptive but does not give enough information and an explanation of why this choice of material has resulted in the best solution.

Response: We thank for the reviewer’s comment, as per the comment all the details are addressed in the revised manuscript.

  1. Table 1 and the context the authors provided are rather coarse. The authors do not adequately explain why the reader of this article should use the present material as well as the present method rather than the other existing materials and protocols that possibly work quite well. I suggest that the authors add more specific features of their work, especially the preparation protocol and steps in comparison to others in Table 1. Besides excellent LOD and wide linear range, more explanation of why the reader should use the protocol presented in this work rather than the others should be discussed in detail within the context.

Response: We thank for the reviewer’s comment, as per the comment the information about this work rather than the others existing material is added in the choice of material and also we added the preparation methods in table 1.

  1. I recommend the authors move the figures and the experimental detail from the supporting material to the main text in the manuscript.

Response: In response to the reviewer as per the comment we moved the data from supplementary information to the revised manuscript.

  1. In Fig.5(a) and (b), the authors only stated the loading amount in volume. The authors should provide the concentration of material used in the text.

Response: We thank for the reviewer’s comment, an average of 0.112 mg cm-2 of the nanocomposite was drop cast on the polished GCE.

  1. For the selectivity, the authors should explain why 4-nitrophenol does not show any effect on furazolidone detection as it also contains the nitro group.

Response: We thank for the reviewer’s comment, and we agree with your point that 4-nitrophenol also contains a nitro group, that is the main reason when compared to other interferences 4-nitrophenol exhibits almost near current response like furazolidone.

  1. The authors mentioned that the calculated EASA of NMO/GCE was higher than bare GCE but the calculated values are not in agreement with the explanation.

Response: We apologize to the reviewer for the mistake, we made all the corrections in the revised manuscript.

  1. Typos and grammatical errors within the manuscript have to be checked.

Response: We apologize to the reviewer for the mistake,  all the typos and grammatical errors corrected in the revised manuscript.

Reviewer 2 Report

In this manuscript Ruspika et al. reported the synthesis of nickel molybdenum oxide nanorods for the electrochemical detection of furazolidone. The synthesized NMO was carefully characterized by XRD, XPS, Raman spectroscopy, FTIR, SEM and EDX. Furthermore, CV and DPV were utilized to detect furazolidone, and testing conditions including scan rate, catalyst loading, electrode, and pH were studied. The result overall is interesting, and the excellent detection limit using NMO is impressive. The synthesized NMO also showed potential for the detection of furazolidone in real samples. Therefore, I would recommend this manuscript to publish after the following points are addressed satisfactorily.

1.       It would be important to review recent progress in the electrochemical detection of furazolidone using nanomaterials.

2.       Figure 1a: It would be better to plot the standard JCPDS data for comparison.

3.       Figure 3: clear scale bars should be provided.

4.       Figure S1e shows the weight percentage not atomic percentage of the material. The corresponding description seems incorrect. In addition, there are other peaks in the EDX spectrum, indicating the presence of impurities. The authors need to discuss it.

5.       The information of NMO electrode preparation needs to be provided.

6.       What do symbols in the Randles-Sevcik equation refer to? Information should be provided.

7.       The authors claimed that NMO/GCE had a higher EASA value, however, the calculated EASA of bare GCE and NMO/GCE was 0.98, and 0.117 cm2, which was inconsistent with the statement.

8.       More experimental details of FRD detection should be given. For example, how was the catalyst prepared and loaded? How was the electrode treated with FRD?

9.       What do legends in Figure 5a refer to? They seem to be the volume of catalyst solution, but the mass loading might be more accurate to describe the contribution of catalyst amount.

10.   Figure 6a: the title of X axis should be “E vs Ag/AgCl (V)” instead of “pH”.

11.   The inset of Figure 8c is difficult to see. A higher resolution figure would be better. Also, a description of the inset is needed in the caption.

Author Response

Reviewer comments and Author’s response

Journal: Micromachines

Manuscript Number:  micromachines-1749731

Authors: Sundaresan Ruspikaa, Vinitha Mariyappana, Shen-Ming Chena,*, Saranvignesh Alagarsamya, Muthumariappan Akilarasana.

Title: Fabrication of a new electrochemical sensor based on bimetal oxide for the detection of furazolidone in biological samples.

Dear Sir/Madam,

We would like to express our sincere gratitude for the valuable suggestions from reviewers to improve the quality of our manuscript. We made all the corrections according to the comments and highlighted them in pink color. The following is the list of amendments and points addressing the comments for your report.

#Reviewer 2

  1. It would be important to review recent progress in the electrochemical detection of furazolidone using nanomaterials.

Response: We thank for the reviewer’s comments, as per your suggestions we included the recent review on the electrochemical detection of furazolidone using nanomaterials in the revised manuscript.

  1. Figure 1a: It would be better to plot the standard JCPDS data for comparison.

Response: We thank for the reviewer’s comments, as per the comment we included the JCPDS data for the comparison plot in fig. 1a in the revised manuscript.

  1. Figure 3: clear scale bars should be provided.

Response: We thank for the reviewer’s comments, as per your suggestions we included the scale bar for fig. 3 in the revised manuscript.

  1. Figure S1e shows the weight percentage not atomic percentage of the material. The corresponding description seems incorrect. In addition, there are other peaks in the EDX spectrum, indicating the presence of impurities. The authors need to discuss it.

Response: We apologize for our mistake we corrected the mistake in the revised manuscript. In response to your question, the other peaks in the EDX spectrum arise due to the indium in which the material was coated for SEM analysis and we cited the reference in the revised manuscript.

  1. The information of NMO electrode preparation needs to be provided.

Response: We thank for the reviewer’s comment, as per the comments we added the information about the fabrication of NMO electrode in the revised manuscript.

  1. What do symbols in the Randles-Sevcik equation refer to? Information should be provided.

Response: We thank for the reviewer’s comment, as per the comments all the corrections have been made in the revised manuscript.

  1. The authors claimed that NMO/GCE had a higher EASA value, however, the calculated EASA of bare GCE and NMO/GCE was 0.98, and 0.117 cm2, which was inconsistent with the statement.

Response: We apologize to the reviewer for the mistake, we made all the corrections in the revised manuscript.

  1. More experimental details of FRD detection should be given. For example, how was the catalyst prepared and loaded? How was the electrode treated with FRD?

Response: We thank for the reviewer’s comment, as per your comments we added the information about the fabrication of NMO electrode and the loading of catalyst in the revised manuscript.

  1. What do legends in Figure 5a refer to? They seem to be the volume of catalyst solution, but the mass loading might be more accurate to describe the contribution of catalyst amount.

Response: We thank for the reviewer’s comment, an average of 0.112 mg cm-2 of the nanocomposite was drop cast on the polished GCE.

  1. Figure 6a: the title of X axis should be “E vs Ag/AgCl (V)” instead of “pH”.

Response: We apologize to the reviewer for the mistake, we made the corrections in the revised manuscript.

  1. The inset of Figure 8c is difficult to see. A higher resolution figure would be better. Also, a description of the inset is needed in the caption.

Response: We thank for the reviewer’s comment, as per the comments all the corrections have been made in the revised manuscript.
